# Brewed Robusta Coffee Increases Nickel Ion Release from Dental Alloys: An *In Vitro* Study

**DOI:** 10.3390/ma14227069

**Published:** 2021-11-21

**Authors:** Dessy Rachmawati, Devanti Ayu Cahyasari, Ardin Tito Febiantama, Lusi Hidayati, Cornelis Johannes Kleverlaan

**Affiliations:** 1Department of Dental Biomedical Science, Faculty of Dentistry, University of Jember, Jl. Kalimantan 37, Jember 68121, East Java, Indonesia; devantiayucahyasari98@gmail.com (D.A.C.); ardintito01@gmail.com (A.T.F.); 2Department of Dental Materials Science, Faculty of Dentistry, University of Jember, Jl. Kalimantan 37, Jember 68121, East Java, Indonesia; hidayatilusi2@gmail.com; 3Department of Dental Materials Science, Academic Centre for Dentistry Amsterdam (ACTA), 1081 LA Amsterdam, The Netherlands; c.kleverlaan@acta.nl

**Keywords:** NiCr, ion release, atomic absorption spectrometry, X-ray fluorescence, scanning electron microscopy

## Abstract

Alloys in oral cavity always interact with dynamic oral environment, such as pH, temperature, salivary conditions, and dietary habits. Coffee can further decrease pH in the oral cavity. Thus, coffee may increase the release of metal ions that may lead to various health diseases. This study aimed to quantitively investigate the effect of brewed Robusta coffee on the nickel ion release and their morphological structures; Methods: 20 alloy specimens were divided into 4 groups and placed in solutions for 48 and 168 h: (1) distilled water, (2) artificial saliva, (3) Robusta coffee, and (4) mixture of artificial saliva and Robusta coffee. AAS, XRF, and SEM were used for examinations; Results: The release of Ni^2+^ was found in all groups and robust release were found in the coffee only and mixture of coffee + artificial saliva solution after 168 h. Likewise, SEM showed that internal oxidation was high after 168 h of immersion; Conclusions: The concentration of nickel increased in saliva of low pH due to brewed coffee. Though the release of these ions is still within the tolerable amount in human body, it should be realized that it usually lasts for months or years. Dentists should be cautious when using instructions for patients with a history of allergies especially for dietary with low acidity.

## 1. Introduction

Metals are still often used for various purposes in dentistry since up to now distinct metals are still irreplaceable by other materials. Moreover, mechanically, and physically, it has the greatest strength to withstand chewing loads. In dentistry, metal is never used in a singular form but always mixed with at least two different types of metals called alloys [1].

Nickel (Ni) is an important transition element essential for many dental alloys. Nickel-based (non-precious) alloys are used in various types of dental restoration (fillings, crowns, bridges, partial dentures) and orthodontic appliances (wires, dental bands, brackets, etc.) [1,2]. Today, nickel is still used in the dental field for a variety of reasons such as corrosion resistance and low costs [3]. Nickel is used in dental construction from a few percent to over 50 percent. However, nickel is the most common metal sensitizer in humans, which leads to the controversy of the long-term use of nickel casting alloys in dentistry [4]. In sensitized individuals, even low doses of nickel (about 6–200 μM) can cause skin inflammation [5]. Exposure to metal ions in the oral cavity can cause adverse reactions, such as inflammatory effects, allergic reactions, and mutagenic effects, depending on the concentration of metal ions released. Immunotoxicity of dental alloy is largely due to the release of Ni^2+^ ions, which can cause several local symptoms such as oral lichen(oid) lessions/oral lichen planus (OLL/OLP) to autoimmune disease (AID) and neurotoxicity [6].

Nickel based alloys in the oral cavity are influenced by temperature, plaque, quality, and quantity of saliva. Likewise, food or drink that is consumed can cause alteration of pH of oral cavity, which can trigger the release of metal ions [1]. The normal pH range of saliva is 6.5–7.0. Of note, protein and chloride ions in saliva affect the release of metal ions. Protein acts as an electrolyte medium that can trigger chemical reactions; meanwhile, chloride has a metal destruction mechanism when they come into contact with the chromium oxide layer which is a protective layer from corrosion in NiCr alloy [7]. Coffee consumption is suggested as one of the activities that can increase the release of metal ions [8].

Coffee is one of the most consumed beverages in the world due to its pleasant taste and aroma [9]. Robusta coffee contains several components such as alkaloid, flavonoid, saponins, tannins, caffeine, and phenol. It has a low pH of around 5.0–5.5 [10] which may cause an increase in the release of metal ions [11]. It can also trigger reduction and oxidation reactions, which lead to increased release of metal ions [12]. The World Health Organization (WHO) guidelines for drinking water currently accept a nickel concentration of 70 μg/mL (10.85 ppm), and the normal range of metal ions in the body is 300–600 μg which is equal to 46.5–93 ppm [13]. Of note, the maximum limit of Ni^2+^ ion in medical applications is <1.10 ppm. However, there are only very limited studies on the corrosion of orthodontic appliances and release of metal ions due to coffee consumption. Some in vivo studies have shown that the level of metal ions in the saliva of patients undergoing orthodontic treatment varies greatly. These changes can be attributed to differences in sample preparation techniques, dietary behavior and different analytical methods. Therefore, in this study, we aimed to analyze in vitro the amount of Ni**^2+^** ion released in the alloy after immersion in Robusta coffee brews to prevent variation. Knowledge on the number of ions released could assist the practitioner in advising an appropriate dietary instruction for the patient’s benefit.

## 2. Materials and Methods

### 2.1. Alloy Specimen Preparation

NiCr dental alloys were selected (Dentecon Inc./Thermabond alloy super cast, MfG, Los Angeles, CA, USA) with the composition of 75% Ni, 15% Cr, 5% Molybdenum (Mo), and 1.6% Beryllium (Be). Each dental alloy was formed into 20 discs, each of which has a diameter of 10 mm and a thickness of 1 mm. All samples were polished with silicon carbide paper (400, 800, 1200 grit) with a grinder polishing machine (M2V; Manfredi). Next, according to [14], samples were cleaned in ethanol and deionized water with an ultrasonic cleaner for 5 min to remove contaminants, and finally autoclaved. Alloys were cast according to the manufacturer’s instructions using the lost-wax technique. All processes of finishing and polishing were carried out in a similar way to simulate the preparation of the cast metal alloys for clinical cases [15,16].

### 2.2. Alloy Immersion in the Solution

Each alloy was placed into a 100 mL clean and dry beaker glass that has been filled with various solutions and incubated at 37 °C to simulate the oral temperature. Next, the pH of each solution was determined by using a pH meter (Hanna Instruments, Hungary). The solutions used included (1) distilled water, (2) artificial saliva, (3) Robusta coffee brews, and (4) mixture of artificial saliva + Robusta coffee. Robusta coffee brew was obtained by dissolving 3 mg of Javanese Robusta coffee powder (PTPN XII, Jember, Indonesia) in a total volume of 100 mL of water. The composition of non-protein added artificial saliva used in this study consisted of 36.0 gr of Sodium Chloride (NaCl); Potassium Chloride (KCl) 1.9 g; Calsium Chloride (CaCl**_2_**) 0.956 g; Sodium bicarbonate (NaHCO**_3_**) 0.85 g; and 400 mL distilled water. After the NiCr alloys were added to each solution, the beaker glasses were wrapped by using aluminum foil. All samples were stored in an incubator at 37 °C for 0, 2 (48 h), and 7 days (168 h). Once every 24 h, the solution containing Robusta coffee was stirred using a glass stirrer.

### 2.3. X-ray Fluorescence (XRF) Analysis

An X-ray Fluorescence (XRF) (Malvern Panalytical Ltd., Malvern WR14 1XZ, United Kingdom) was used to determine the composition of the sample before and after immersion. The following are the test steps using XRF: (1) Prepare a disk-shaped alloy sample that has been immersed, and (2) follow the Standard Operational Procedure (Crocker Nuclear) (Laboratory, 2015).

### 2.4. Atomic Absorption Spectrometry (AAS) Analysis

The total nickel concentration in the solution sample was analyzed by flame atomic absorption spectrometry Flame-AAS (Thermo Scientific iCE 3000 Series, Thermo Fisher Scientific, Cambridge, UK). First, we diluted the solution sample with ultrapure water to increase the volume of the solution. All analyses were based on three repeated readings of each solution sample, and quality control samples of known concentrations were also analyzed. All results given were based on the mean and standard deviations of at least triplicate samples for each alloy specimens, solution, and time period, and subtracted to the respective blank sample concentration. The detection limit of each element was calculated 0.10 as 3 times the SD of the blank (3 s blank, *n* = 5 specimens); following the Graphite Furnace (GF) -AAS method, the detection limit for Ni 0.05 was calculated as 0.003 parts per billion (ppb).

### 2.5. Scanning Electron Microscopy (SEM) Analysis

Next, the surface morphology of NiCr alloy before and after immersion was measured using SEM analysis (Hitachi TM3000, Hitachi, Tokyo, Japan). The measurement were in the form of the percentage of ions and oxides. In addition, the surface morphology of the NiCr alloy was observed at a magnification of 2500×.

### 2.6. Data Analysis

The statistical significance of the effects of Robusta coffee brews on the Ni^2+^ ion release was analyzed by using two-way ANOVA and Dunnett test (parametric ANOVA), with statistic program GraphPad Prism Software version 9.0 (San Diego, CA, USA). *p* ≤ 0.05 was considered statistically significant. All data are presented as mean ± SD.

## 3. Results

### 3.1. The Composition of Nickel Chromium Alloys

First, to quantitatively explore the effect of Robusta coffee brews on the Ni^2+^ ion release and their morphological structures, XRF was conducted to calculate the original composition of metal ion in gram per amount (wt%). The results of the purity of NiCr alloy (pre-test) can be seen in Table 1. The highest percentage of ions is Ni 75%, followed by chromium 15%. Whilst the percentage of Ni^2+^ ion after immersion can be seen in Table 2 which shows that several treatments resulted in a lower percentage of Ni than the pretested alloys. These results confirm that the alloy used is a NiCr alloy, of which nickel is the main composition. The data in Table 2 indicate that there are differences in the composition of the alloy specimens before and after immersion. It also shows that the more Ni^2+^ ions are released, the composition in the NiCr alloy also decreases.

### 3.2. pH Meter Result

The solutions-pH results are shown in Table 3. Test results are included (1) distilled water pH of 7.2, (2) artificial saliva pH of 6.5, (3) Robusta coffee brews pH of 5.0, and (4) mixture of artificial saliva + Robusta coffee pH of 5.5. Knowledge of the results obtained in this section is useful in supporting the hypothesis that the release of Ni^2+^ ions is thought to be caused by the low acidity of the coffee.

### 3.3. Nickel Ion Released according to Atomic Absorption Spectrophotometry (AAS)

Subsequently, Nickel ion released according to Atomic Absorption Spectrometry (AAS) is summarized in Figure 1, Table 4 Amongst 48 h immersion of NiCr + (artificial saliva + Robusta coffee) showed the highest concentration of Ni 4.26 ppb. It gradually increased and was statistically significant in all groups of 168 h of immersion. A bar chart of Ni^2+^ ion release can be seen in Figure 1 and in numerical Table 4.

### 3.4. The Surface Morphology of Nickel Chromium Alloys

The next phase carried out was Scanning Electron Microscopy (SEM). This process resulted in the before and after immersion surface morphology of NiCr alloys which can be seen in Figure 2 below. The arrows show the parts of the alloy which have undergone internal oxidation, thus forming a pore.

The results indicate that the positive control group immersed in artificial saliva (2C) with 4.24 ppb and Robusta coffee brew after 168 h (2D) showed the release of 5.67 ppb Ni^2+^ ions which can cause significant changes in the surface morphology of the alloy. SEM results of NiCr alloys that have been immersed in various immersion solutions showed different surface roughness, pore sizes, and pore numbers. In Figure 2E, the alloy surface morphology which has been immersed in the treatment solution (artificial saliva + coffee) appears coarser with higher number of pores than the others. In Figure 2A the pretest alloys and (B) the alloys that have been soaked in distilled water show smoother alloy surface morphology, and there are no oxidized surfaces. The level of surface roughness is directly proportional to the results of the AAS, except in the distilled water group. A higher ion release concentration indicates a higher degree of surface roughness.

The SEM results conclude that the treatment group (saliva + coffee) has the highest roughness level; the positive control group (NiCr + Artificial Saliva; NiCr + Coffee) generates a moderate roughness level; the negative control group (NiCr + distilled water) and the pretest have the lowest roughness level.

## 4. Discussion

The release of nickel ions in oral cavity is caused by the electrolyte content in artificial saliva. One of the electrolytes that play a role in releasing ions is chloride ion. Chloride can damage the oxide layer on the alloy surface, causing the release of ions [17]. Artificial saliva contains organic components in the form of proteins which can act as an electrolyte medium, thus triggering chemical reactions. In addition, nickel ions are more likely to detach due to the structure of their atomic elements. Therefore, the release of nickel ions in artificial saliva immersion is quite high [18].

Figure 1 shows the increasing release of Ni^2+^ ions in the immersion solution of pure Robusta coffee. Correspondingly, as seen from the low pH of Robusta coffee compared to other soaking solutions (Table 3), the results of nickel ions release are in line with [19] who state that in acidic environments the number of H**^+^** ions will increase causing them to become corrosive, thus the ions are easily released. Next, when compared to the results of combining artificial saliva + Robusta coffee on 48 and 168 h, the amount of nickel ion released is even higher. In fact, the pH in pure Robusta coffee immersion is the lowest among other solutions (pH 5), while the pH of the Robusta + saliva coffee mixture is 5.5. Presumably, this may result from the corrosion rate inhibitory properties in the alloy of one of the ingredients of pure Robusta coffee, namely concentrated caffeine. Caffeine (C**_8_**H**_10_**N**_4_**O**_2_**) is a compound containing an N group which has free electrons. The N group will donate its free electrons to the alloy, thus inhibiting the corrosion rate in the immersion of pure coffee [20]. According to [21], the high concentration of caffeine in a solution will reduce the corrosion rate of an alloy. This is in accordance with the results of their study which evinces that applying a concentration of 2% caffeine results in a greater reduction in corrosion rates compared to the application of 1% caffeine over time.

In addition, the antioxidant content in Robusta coffee can also affect the release of ions. Antioxidants are compounds that have an inhibition mechanism toward oxidation reactions, by binding to free ions and highly reactive molecules. The antioxidants in Robusta coffee beans include the tannins, flavonoids, and polyphenols. Tannins and flavonoids have properties as chelating agents, which means they can bind metal ions [19,21]. Therefore, the results of this study indicate that the release of ions in pure Robusta coffee is not higher, despite acidic pH, compared to the mixture of coffee and artificial saliva, which reflects the state of brewing coffee when consumed and mixed with saliva in the oral cavity. The high release of nickel ions is caused by the interaction between artificial saliva and Robusta coffee, which results in a decrease in buffer function. The decrease in buffer function distracts saliva from maintaining its pH balance, therefore it cannot bind H^+^. In addition, the carbohydrate content in coffee can be fermented by acid producing bacteria; thus, reducing pH in saliva [22]. Therefore, the release of nickel ions in the mixture of Robusta coffee + artificial saliva group is the highest.

The release of metal ions is not only influenced by pH, but is also influenced by the type of alloy, manufacturing process, surface area, immersion time, and the composition of the drink/food consumed [8,23]. Although there is no literature investigating the release of Ni^2+^ ions in vitro due to coffee immersion as a comparison, several research results show that soaking orthodontic braces with several types of carbonated drinks and orange juice shows an increase in the release of Ni^2+^ ions [24]. In line with this study, Ref. [8] show that consumption of food containing low pH (such as fruit juices, coffee, yogurt, and vinegar) can change the acidity of oral cavity therefore it can also increase the release of Ni^2+^ ions from the use of orthodontic devices used. There are also in vivo references to patients who state that the increase in Ni^2+^ ions release in coffee-consuming patients is approximately 1.22× higher than in those who do not consume coffee [8].

The interaction between saliva + Robusta coffee causes an increase in the release of Ni^2+^ ions and this sample even shows the highest release. The high Ni^2+^ ion release stems from the influence of saliva content. Saliva consists of various electrolytes, such as sodium, potassium, calcium, magnesium, bicarbonate, and phosphate. In addition, saliva also contains immunoglobulins, enzymes, mucin, other proteins, and several products such as urea and ammonia [10]. In this study, the artificial saliva contained components of potassium, calcium, bicarbonate, and chloride. These are electrolytes that can trigger electrochemical reactions. An electrochemical reaction is the reaction between anode and cathode. Anode is associated with an oxidation reaction (releasing electrons), and cathode is associated with a reduction reaction (accepting electrons). In this case, metal ions serve as the anode, and H^+^ ions from the electrolyte medium serve as the cathode. As a corollary, an electrochemical reaction occurs, which causes the release of ions in the alloy [25]. Therefore, the release of nickel ions in the immersion test involving artificial saliva + coffee was found the highest.

The XRF test results in Table 2 shows that there is a correspondence between the AAS and XRF test results. This is in line with the theory which states that the group found to have the highest amount of nickel ion release tends to have the lowest nickel composition. This was shown by the XRF test data in this study, showing that the percentage of nickel in the alloy after immersion was lower than that in the pre-test alloy with a nickel percentage of 75%. Meanwhile the alloy immersed for 168 h immersion in a mixture of saliva + Robusta coffee had a nickel percentage of 67.39%. In addition, the XRF test results confirmed that there were differences in the composition of the alloys before and after immersion.

SEM was conducted to determine the surface roughness and porosity of the NiCr alloy. Surface roughness is the irregularity of a surface in the form of curves or scratches [20]. Based on Figure 2 shows that different alloys display different surface roughness, number of pores, and pore sizes. To date, there is no literature on the amount of minimal Ni^2+^ ion levels that can cause changes in the surface morphology of an alloy. The results of this study, the Ni^2+^ ion content of 1.4 ppb in the NiCr alloy immersed in distilled water underwent changes in surface morphology.

The SEM assays were observed in the 168 h immersion. The findings demonstrated that longer immersion time leads to more Ni^2+^ ions released. Therefore, the alloy surface morphological structure showed the difference between pre-test samples and those in several types of immersion: distilled water, artificial saliva, coffee, and mixtures (artificial saliva + Robusta coffee). SEM test results showed that the pre-test and NiCr alloys immersed in distilled water had the lowest surface roughness level. Meanwhile, NiCr immersed in artificial saliva had medium surface roughness, the alloy surface roughness increases, such as what was found in NiCr immersed in coffee. In addition, the mixture of Robusta coffee and artificial saliva resulted in the highest surface roughness. SEM test results were directly proportional to AAS test results. The high release of nickel ions will indicate a higher level of surface roughness and a higher number of pores. In all types of immersion, black nodules were found on the surface of the NiCr alloy caused by internal oxidation of less noble alloy elements, such as copper and zinc [25]. Black nodules were also more clearly seen in NiCr + artificial saliva, NiCr + coffee and NiCr + (Saliva + Robusta coffee). This was due to the pH of the solution with potent acidity, causing more alloy parts to be oxidized.

The release of nickel ions in all groups after 168 h immersion was higher than 48 h. According to ASTM International (2019), the standard immersion time is often used is from 48 to 168 h. Immersion time which is less than 48 h will result in a high rate of ion release. Subsequently, the ion release rate will drop/stabilize after 168 h. Furthermore, according to [26], long immersion times can lead to increased ion release associated with the metal’s susceptibility to corrosion. Of note, although the release of metal ions may be negligible, it should be realized that it usually lasts for months or years. Likewise, our previous results showed that the optimal effective innate stimulatory dosages range from 125–750 µM for Ni^2+^. As a result, the levels of metal ion released in the oral cavity and thereafter absorbed in the gastrointestinal system are extremely negligible [25,26]. Nonetheless, several secondary factors may promote the development of local and systemic complaints [1].

A more detailed examination of the influence of coffee consumption on the release of metal ions from dental appliances will be useful to allow nutritional recommendations for patients. These recommendations will reduce patient exposure to nickel released from dental appliances.

## 5. Conclusions

Coffee consumption can increase the release of Ni^2+^ ions from metal constructions in the oral cavity. Immersion of NiCr + (artificial saliva + Robusta coffee) showed the highest concentration of Ni^2+^ with 4.26 ppb (in 48 h) and 6.77 ppb (in 168 h). Even though the release of these ions is still within the tolerable amount in the human body, dentists should be cautious when giving instructions to patients with a history of allergies. Therefore, it may be useful to recommend dental patients to limit the consumption of food and beverages characterized by low pH to reduce the number of ions dissolved from the dental alloys.

## Figures and Tables

**Figure 1 materials-14-07069-f001:**
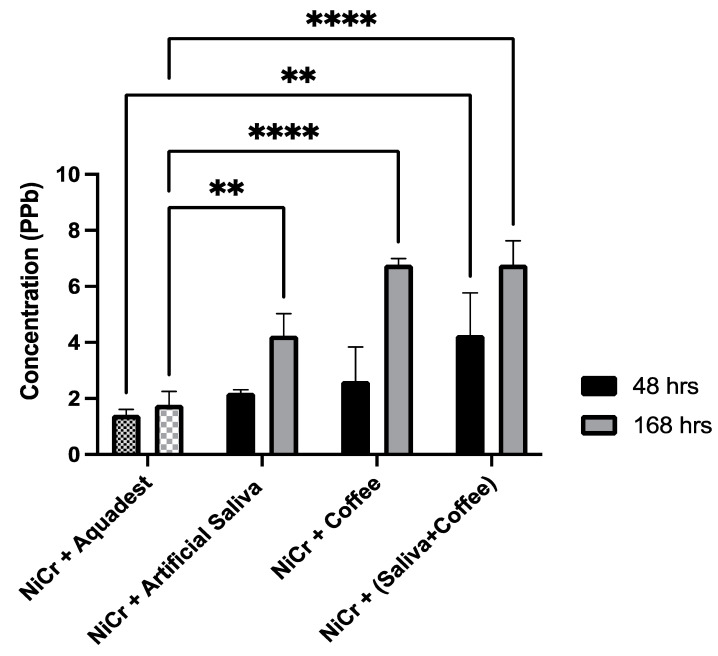
Ni^2+^ ion release from NiCr alloys after immersed in various solutions after 48 and 168 h (in ppb) according to Atomic Absorption Spectrometry (AAS). Graphical bars (Figure 1) and concentration in numerical (Table 4), the data represent mean ± SD from five specimens (*n* = 5). Asterisks specify statistically significant (Two-way ANOVA and Dunnett test (parametric ANOVA)) differences in Ni^2+^ ion release from each solution as compared to the control NiCr + distilled water, ** *p* < 0.01, **** *p* < 0.001.

**Figure 2 materials-14-07069-f002:**
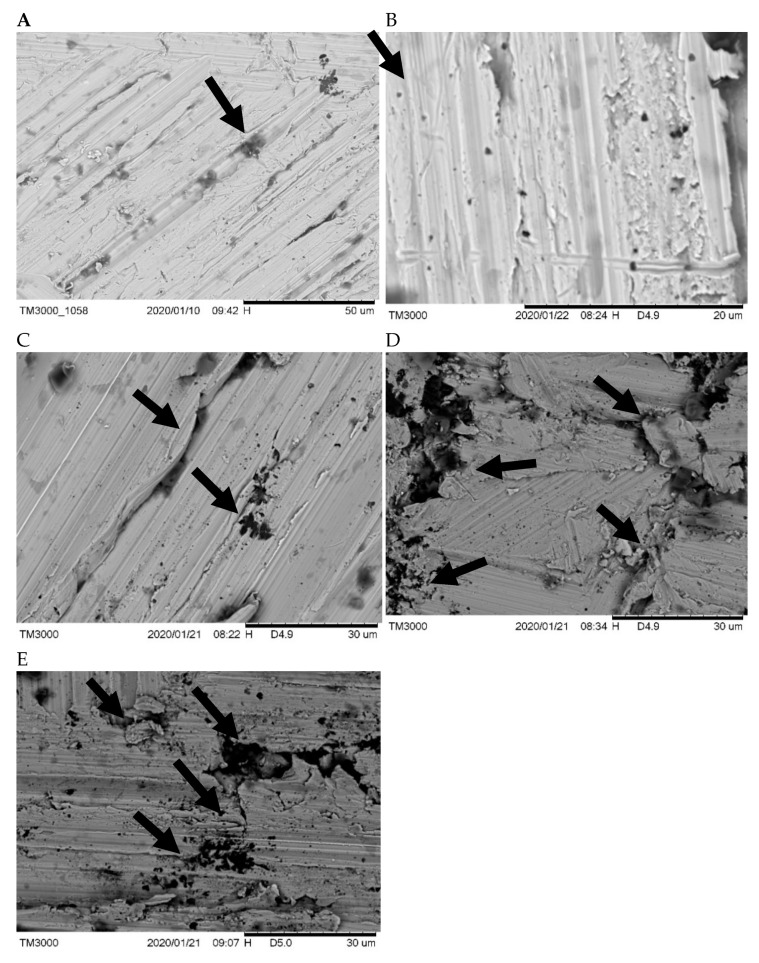
Scanning Electron Microscopy (SEM) 2500x-morfological structure images of alloy NiCr before and after immersed in various solutions for 168 h (respective image). Scanning Electron Microscopy images of morphological structure of alloy NiCr after immersed in solutions: (**A**) NiCr pre-test (as received), (**B**) NiCr + distilled water, (**C**) NiCr + Artificial Saliva, (**D**) NiCr + Coffee, (**E**) NiCr + (artificial saliva + coffee).

**Table 1 materials-14-07069-t001:** Composition Alloy NiCr Pretest (in wt%) according to XRF test (*n* = 5).

	Component	Composition (in wt%) ± SD
Alloy NiCr	Nickel (Ni^2+^) ion	75% ± 0.46
	Chromium (Cr) ion	15% ± 1.13
	Molybdenum (Mo) ion	5% ± 1.27
	Phosphor (P) ion	3.0% ± 0.98
	Others (Ca, Sc, Ti, Mn, Fe, Co, Y, Zr, Ba, Er, Au, Th)	4.5% ± 0.67

**Table 2 materials-14-07069-t002:** Composition alloy NiCr (in wt%) according to XRF test after 48 and 168 h immersion (*n* = 5).

		Composition (in wt%) ± SD	
pH		48 h	168 h
7.2	NiCr Pre-test NiCr alloy + Distilled water	75% ± 1.25 74.7% ± 0.8	75% ± 0.76 74.52% ± 0.95
6.5	NiCr alloy + Artificial saliva	73.8% ± 0.5	71.7% ± 0.43
5.0	NiCr alloy + Robusta coffee	72.4% ± 1.1	69.02% ± 0.92
5.5	NiCr alloy + Artificial saliva + Robusta coffee	70.48% ± 0.85	67.39% ± 0.24

**Table 3 materials-14-07069-t003:** Average pH solutions in four groups (*n* = 5 in each group), according to pH meter measurements. Results represent mean ± SD.

Group	pH ± SD
NiCr alloy + Distilled water	7.2 ± 0.05
NiCr alloy + Artificial saliva	6.5 ± 1.15
NiCr alloy + Robusta Coffee	5.0 ± 0.08
NiCr alloy + Artificial saliva + Robusta Coffee	5.5 ± 1.5

**Table 4 materials-14-07069-t004:** Concentration of Ni (in ppb) in solution according to AAS test after 48 an168 h of immersion.

pH	Group	Concentration (ppb)	
		48 h	168 h
7.2	NiCr Pre-test NiCr alloy + Aquadest	0.03 ± 0.1 1.41 ± 0.2	0.05 ± 0.05 1.77 ± 0.48
6.5	NiCr alloy + Artificial saliva	2.20 ± 0.11	4.24± 0.79 **
5.0	NiCr alloy + Robusta Coffee	2.62 ± 1.22	5.67 ± 0.23 ****
5.5	NiCr alloy + Artificial saliva + Robusta Coffee	4.26 ± 1.50 **	6.77 ± 0.85 ****

** *p* < 0.01 and **** *p* < 0.001.

## Data Availability

Not applicable.

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
