# Peer review of "Brewed Robusta Coffee Increases Nickel Ion Release from Dental Alloys: An In Vitro Study"

_materials, 2021, doi:10.3390/ma14227069_

Round 1

Reviewer 1 Report

Abstract must be enriched via valuable results which pave the way for understanding the audiences.

Introduction. The analytical techniques used for the determination of metal ions release from dental alloys should be presented. Please elaborate. Lines 65-67 should detail “the in vivo studies”…. There is no explanation of significant novelty of the work as compared with their previous work. This should be highlighted and compared with their previous work. What are the new findings in this work and how it differs from their previous work?

Results and Discussion. Please replace “Ni ion” with “Ni2+ ion. Also, to increase the scientific value of the manuscript Authors should consider extension of the whole results section with comparison of obtained results with the results described in previous publications.

The conclusion section should have the main results in quantitative statements as well. A good conclusion shows the principle topic, key arguments and might suggest future research.

The results obtained are interesting and promising. The manuscript can be accepted for publication in Materials journal only after MAJOR corrections. If the paper will not be significant improved, I will not recommend the publication.

Author Response

Manuscript must be enriched via valuable results which pave the way for understanding the audiences.

We have changed and enriched the manuscript via valuable results as suggested.

Introduction. The analytical techniques used for the determination of metal ions release from dental alloys should be presented. Please elaborate. Lines 65-67 should detail “the in vivo studies”…. There is no explanation of significant novelty of the work as compared with their previous work. This should be highlighted and compared with their previous work. What are the new findings in this work and how it differs from their previous work?

We have changed the introduction as suggested. According to our understanding and based on available publications, this is the first experiment which is showing the complete result on how the NiCr alloys ion act if they are immersed in the brewed coffee. In the manuscript, we referred to other references available.

Results and Discussion. Please replace “Ni ion” with “Ni2+ ion. Also, to increase the scientific value of the manuscript Authors should consider extension of the whole results section with comparison of obtained results with the results described in previous publications.

We have replaced the Ni ion into Ni2+ ion as suggested, and have also added comparison of obtained results with results as described in previous publications.

The conclusion section should have the main results in quantitative statements as well. A good conclusion shows the principle topic, key arguments and might suggest future research.

We have changed the conclusion section as suggested.

The results obtained are interesting and promising. The manuscript can be accepted for publication in Materials journal only after MAJOR corrections. If the paper will not be significant improved, I will not recommend the publication.

Thanks to the reviewer for the observations and remarks!

Reviewer 2 Report

The manuscript is strong regarding the data on less studied release of metals from dental alloys. The manuscript regarding the topic and results presented is of interest to the scientific community and revisions based on the comments below are recommended before considering for publication.

Major comments:

The data is very descriptive and not discussed in detail, examples are: line 122-, 130-, 144-, 158-. In addition, the discussion section is not reflecting the data and the manuscript.  

Lack of  literature to support the data or discussion.....

Writing format: Sometimes, two or three conjunctions are used in one sentence, which har to follow, e.g. first, next, consequently.....  All the abbreviations are not introduced, even when they were introduced the long names were used instead of the abbreviations (I highlighted some of them).

The English language is hard to follow sometime, e.g. line; 110, 175, 182,  214....

In conclusion, I would suggest to conclude the main finding or discuss the impact of the outcome. The current version of the conclusion is primarily repeating the result. Please consider rewriting it.

For minor comments, please see the attached file

Author Response

The manuscript is strong regarding the data on less studied release of metals from dental alloys. The manuscript regarding the topic and results presented is of interest to the scientific community and revisions based on the comments below are recommended before considering for publication.

Thanks to the reviewer for the observations and remarks!

Major comments:

The data is very descriptive and not discussed in detail, examples are: line 122-, 130-, 144-, 158-. In addition, the discussion section is not reflecting the data and the manuscript.  

We have explained more details in the data, as well as in the discussion section to reflect the data and manuscript.

Lack of literature to support the data or discussion.

We have added literature to support the data.

Writing format: Sometimes, two or three conjunctions are used in one sentence, which hard to follow, e.g. first, next, consequently.....  All the abbreviations are not introduced, even when they were introduced the long names were used instead of the abbreviations (I highlighted some of them).

We have made several rewriting changes to make the manuscript more easy to read. Before each abbreviation, we have used the complete word. We have also made a list of all used abbreviations. 

The English language is hard to follow sometime, e.g. line; 110, 175, 182,  214....

We have changed some sentences to make the manuscript more easy to read and understand.

In conclusion, I would suggest to conclude the main finding or discuss the impact of the outcome. The current version of the conclusion is primarily repeating the result. Please consider rewriting it.

We now have changed the conclusion as suggested.

For minor comments, please see the attached file

Not convinced yet how the pH affect the release of metal. Needs more elaboration! (Question for line 56-59 from the attached PDF)

We have made a more detailed description. Corrosion is an electrochemical reaction on the metal surface, which leads to the release of ions by the metal. This process can occur as a result of internal and external factors. The internal factors affecting corrosion include metal composition and structure. On the other hand, the external factors that could affect corrosion are the biological environment, pH and temperature.

A lower pH can affect the ion release of appliances and alloys in the oral cavity. Metal is released into the oral cavity with saliva as the medium. This could be influenced by a high chloride mixture in the saliva or the intake of various foods and drinks with low pH levels.

Reviewer 3 Report

This is a very well designed study on the effects of coffee on Ni ions release from different alloys used in dental applications.

Major point:
In the Discussion (4., lines 170-171) it is said that "Artificial saliva contains organic components in the form of proteins". However, in the Methods section (2.2, lines 86-88) the artificial saliva does not include proteins of any kind. Please correct accordingly or mention that the artificial saliva used in the study did not include proteins.

Minor points:
Change 'subconssious' in lines 24 and 237 to "negligible';
Define OLL/OLP (line 47);
Italicise "in vitro" and "in vivo" in lines 65, 68, 208, 213.
Change "mimicked" (l. 83) to "mimick";Change the segment (l. 38-39) "...variety of reason such as corrosion resistant..." to "variety of reasons such as corrosion resistance";
Change subtitle in line 81 to "Alloy immersion in the solution";
Change segment in lines 93-94 "To determine the composition of the sample before and after immersion was done using an X-Ray Fluorescence (XRF)" to " To determine the composition of the sample before and after immersion an X-Ray Fluorescence (XRF) assay was performed";
Change "treatment" (l. 126) to "treatments";
Capitalize "Robusta" in all occurrences (l. 131, 132, 272, 273);
Reorganize the segment (l. 220-221) "In addition, saliva also contains immunoglobulins, proteins, enzymes, mucin and several products such as urea and ammonia" to "In addition, saliva also contains immunoglobulins, enzymes, mucin, other proteins, and several products such as urea and ammonia"

Author Response

This is a very well designed study on the effects of coffee on Ni ions release from different alloys used in dental applications.

Thanks to the reviewer for the observations and remarks!

Major point:
In the Discussion (4., lines 170-171) it is said that "Artificial saliva contains organic components in the form of proteins". However, in the Methods section (2.2, lines 86-88) the artificial saliva does not include proteins of any kind. Please correct accordingly or mention that the artificial saliva used in the study did not include proteins.

We have corrected this point of discussion in the manuscript.

Minor points:

Change 'subconssious' in lines 24 and 237 to "negligible'; Define OLL/OLP (line 47); Italicise "in vitro" and "in vivo" in lines 65, 68, 208, 213.

Change "mimicked" (l. 83) to "mimick";Change the segment (l. 38-39) "...variety of reason such as corrosion resistant..." to "variety of reasons such as corrosion resistance";

Change subtitle in line 81 to "Alloy immersion in the solution";

Change segment in lines 93-94 "To determine the composition of the sample before and after immersion was done using an X-Ray Fluorescence (XRF)" to " To determine the composition of the sample before and after immersion an X-Ray Fluorescence (XRF) assay was performed";

Change "treatment" (l. 126) to "treatments"; Capitalize "Robusta" in all occurrences (l. 131, 132, 272, 273);

Reorganize the segment (l. 220-221) "In addition, saliva also contains immunoglobulins, proteins, enzymes, mucin and several products such as urea and ammonia" to "In addition, saliva also contains immunoglobulins, enzymes, mucin, other proteins, and several products such as urea and ammonia".

We have made all changes as requested for these minor points.

Round 2

Reviewer 1 Report

The conclusion section should have the main results in quantitative statements as well. 

Author Response

We have added the conclusion section as suggested.

Reviewer 2 Report

The authors address most of my comments. Therefore, I recommend the manuscript for publication.

Author Response

Thanks to the reviewer for the observations and remarks!